# Role of SIRT-3, p-mTOR and HIF-1α in Hepatocellular Carcinoma Patients Affected by Metabolic Dysfunctions and in Chronic Treatment with Metformin

**DOI:** 10.3390/ijms20061503

**Published:** 2019-03-26

**Authors:** Serena De Matteis, Emanuela Scarpi, Anna Maria Granato, Umberto Vespasiani-Gentilucci, Giuliano La Barba, Francesco Giuseppe Foschi, Erika Bandini, Martina Ghetti, Giorgia Marisi, Paola Cravero, Laura Gramantieri, Alessandro Cucchetti, Giorgio Ercolani, Daniele Santini, Giovanni Luca Frassineti, Luca Faloppi, Mario Scartozzi, Stefano Cascinu, Andrea Casadei-Gardini

**Affiliations:** 1Biosciences Laboratory, Istituto Scientifico Romagnolo per lo Studio e la Cura dei Tumori (IRST) IRCCS, 47014 Meldola, Italy; erika.bandini@irst.emr.it (E.B.); martina.ghetti@irst.emr.it (M.G.); giorgia.marisi@irst.emr.it (G.M.); 2Biostatistics and Clinical Trials Unit, Istituto Scientifico Romagnolo per lo Studio e la Cura dei Tumori (IRST) IRCCS, 47014 Meldola, Italy; emanuela.scarpi@irst.emr.it; 3Immunotherapy Unit, Istituto Scientifico Romagnolo per lo Studio e Cura dei Tumori (IRST) IRCCS, 47014 Meldola, Italy; annamaria.granato@irst.emr.it; 4Interdisciplinary Center for Biomedical Research (CIR), Laboratory of Internal Medicine and Hepatology, Campus Bio-Medico University, 00128 Rome, Italy; U.Vespasiani@unicampus.it; 5Department of General Surgery, Morgagni-Pierantoni Hospital, 47121 Forlì, Italy; giuliano.labarba@auslromagna.it; 6Department of Internal Medicine, Degli Infermi Hospital, 48018 Faenza, Italy; francesco.foschi@auslromagna.it; 7Department of Medical Oncology, Istituto Scientifico Romagnolo per lo Studio e Cura dei Tumori (IRST) IRCCS, 47014 Meldola, Italy; paola.cravero@irst.emr.it (P.C.); luca.frassineti@irst.emr.it (G.L.F.); 8Center for Applied Biomedical Research (CRBa), St. Orsola-Malpighi University Hospital, 40138 Bologna, Italy; laura.gramantieri@aosp.bo.it; 9Department of Medical and Surgical Sciences, Alma Mater Studiorum, University of Bologna, 40126 Bologna, Italy; alessandro.cucchett2@unibo.it (A.C.); giorgio.ercolani@auslromagna.it (G.E.); 10Medical Oncology Unit, Campus Biomedico University, 00128 Rome, Italy; D.Santini@unicampus.it; 11Medical Oncology, University Hospital of Cagliari, 09124 Cagliari, Italy; lucafaloppi@gmail.com (L.F.); marioscartozzi@gmail.com (M.S.); 12Division of Medical Oncology, Department of Medical and Surgical Sciences for Children and Adults, University Hospital of Modena, 41122 Modena, Italy; cascinu@yahoo.com (S.C.); casadeigardini@gmail.com (A.C.-G.)

**Keywords:** non-alcoholic steatohepatitis, metabolic syndrome, diabetes, metformin

## Abstract

The incidence of hepatocellular carcinoma deriving from metabolic dysfunctions has increased in the last years. Sirtuin- (SIRT-3), phospho-mammalian target of rapamycin (p-mTOR) and hypoxia-inducible factor- (HIF-1α) are involved in metabolism and cancer. However, their role in hepatocellular carcinoma (HCC) metabolism, drug resistance and progression remains unclear. This study aimed to better clarify the biological and clinical function of these markers in HCC patients, in relation to the presence of metabolic alterations, metformin therapy and clinical outcome. A total of 70 HCC patients were enrolled: 48 and 22 of whom were in early stage and advanced stage, respectively. The expression levels of the three markers were assessed by immunohistochemistry and summarized using descriptive statistics. SIRT-3 expression was higher in diabetic than non-diabetic patients, and in metformin-treated than insulin-treated patients. Interestingly, p-mTOR was higher in patients with metabolic syndrome than those with different etiology, and, similar to SIRT-3, in metformin-treated than insulin-treated patients. Moreover, our results describe a slight, albeit not significant, benefit of high SIRT-3 and a significant benefit of high nuclear HIF-1α expression in early-stage patients, whereas high levels of p-mTOR correlated with worse prognosis in advanced-stage patients. Our study highlighted the involvement of SIRT-3 and p-mTOR in metabolic dysfunctions that occur in HCC patients, and suggested SIRT-3 and HIF-1α as predictors of prognosis in early-stage HCC patients, and p-mTOR as target for the treatment of advanced-stage HCC.

## 1. Introduction

Hepatocellular carcinoma (HCC) represents the fifth most common cancer worldwide and the second most frequent cause of death among cancers [1]. The incidence of HCC cases deriving from metabolic diseases such as metabolic syndrome and non-alcoholic steatohepatitis (NASH) has increased [2].

In our recent review, we provided insight into the involvement of sirtuin-3 (SIRT-3), a member of the mitochondrial sirtuin family, in metabolism dysfunctions, suggesting its putative bridge role between metabolic diseases and HCC [3]. Furthermore, our research group published a work showing high expression of SIRT-3 in HCC patients with type-2 diabetes mellitus (T2DM) and in those undergoing chronic treatment with metformin [4].

Casadei-Gardini et al. showed a lower response to sorafenib in HCC patients undergoing chronic therapy with metformin for T2DM in respect with non-diabetic patients or those taking insulin [5], and suggested an involvement of SIRT-3 in the mechanism of resistance to sorafenib in this setting of patients [4].

It is known that metformin can downregulate mammalian target of rapamycin (mTOR) directly by activation of AMP-activated protein kinase (AMPK), which, in turn, positively regulates SIRT-3 or indirectly not involving the AMPK/SIRT-3 pathway [6,7,8].

The role of SIRT-3 and its downstream effectors including HIF-1α and p-mTOR in HCC metabolism remains unclear. There is a controversy whether these proteins act as oncogenes or suppressor genes in tumors [9,10,11], although there are literature data about a tumor-suppressive function of SIRT-3 and an oncogenic role of HIF-1α and p-mTOR in HCC [12,13,14,15].

To better clarify the biological and clinical role of SIRT-3, p-mTOR and HIF-1α in HCC, we evaluated the expression of these three markers in tissue from two cohorts of HCC patients (one at early stage and one at advanced stage) in relation with the presence of metabolic dysfunctions, chronic treatment with metformin and clinical outcome. In addition, the effect of metformin alone and in combination with sorafenib has been also explored in three HCC cell lines to better understand the implication of SIRT-3 and its downstream effectors in the drug resistance mechanism.

## 2. Results

### 2.1. Patient Characteristics

Seventy patients were included in the biological study and the clinical characteristics of patients are summarized in Table 1.

Forty-eight patients (40 males and 8 females) with HCC at an early stage were included in the biological study. Median age was 70 years (IQR 65–77 years). The most common etiologies of liver disease were viral infection (41.6%), metabolic liver disease (29.2%) and other (29.2%). Twenty (42.5%) patients had T2DM, of whom 15 (31.9%) were treated with metformin. Twenty-two patients (20 males and 2 females) with HCC at an advanced stage were also included in this study. Median age was 72 years (IQR 63–80 years). The common etiologies were viral infection (50%), metabolic syndrome (40.9%) and other (9.1%). Of these, 11 (50%) patients had T2DM and 8 (36.4%) patients were treated with metformin. All advanced-stage patients were treated with sorafenib.

### 2.2. Immunohistochemical Expression Pattern of SIRT-3, p-mTOR and Nuclear HIF-1α in Tumor Tissues

We used immunohistochemistry to analyze the expression of different markers in two settings of HCC patients. A representative case related to negative/low and positive expression of the markers in the two populations is shown in Figure 1.

The median SIRT-3, p-mTOR and nuclear HIF-1α expression levels (percent of immunopositive tumor cells) were 35 (IQR 10–60), 0 (IQR 0–0) and 1 (IQR 0–10) in early-stage patients vs. 60 (IQR 10–90), 0 (IQR 0–0) and 0 (IQR 0–7.5) in advanced-stage patients, respectively.

Staining intensity of the SIRT-3 and p-mTOR expression is described in Appendix A for early- and advanced-stage HCC patients. The staining intensity for HIF-1α is not reported because we observed a strong nuclear expression in all positive cases without cytoplasmic expression.

There was no significant difference in the expression levels of all markers between the two cohorts of HCC patients.

The expression of three markers related to etiology (viral vs. metabolic syndrome vs. other), presence of T2DM and diabetes therapy are reported in Figure 2A–C.

We observed a higher expression of SIRT-3 in patients with diabetes (median value of 60%) than non-diabetic patients (median value of 30%) (*p* = 0.011) (Figure 2B), and in patients treated with metformin than those taking insulin (70% vs. 30%, respectively) (*p* = 0.030), as shown in Figure 2C.

Interestingly, p-mTOR resulted more expressed in patients with metabolic syndrome (median value of 0% with a range of positivity in the neoplastic population of 0–100%) than in those with different etiology (*p* = 0.036) (Figure 2A), and in diabetic patients treated with metformin than those taking insulin (median value of 0% with a range from 0% to 100% vs. 0% with a range from 0% to 40%) (*p* = 0.021) (Figure 2C). No significant correlation was observed for HIF-1α.

There was no difference among SIRT-3, p-mTOR and HIF-1α expression levels and other clinical parameters reported in Table 1.

We also evaluated the expression of SIRT-3 and p-mTOR in non-cancerous adjacent liver tissue of HCC patients with diabetes and/or metabolic syndrome, to verify whether their positive expression was limited to cancerous tissue. For the majority of cases, non-cancerous adjacent liver tissue was not present. As reported in Appendix A, we observed a slight positivity of SIRT-3 in two of nine diabetic patients with metabolic syndrome and in four of eight patients with only diabetes. The expression of p-mTOR resulted negative in all cases.

### 2.3. In Vitro Effect of Metformin and Sorafenib on HCC Cell Lines

To better understand the data obtained by the ex vivo study discussed above, we elucidated the effect of metformin on SIRT-3 and p-mTOR protein expression in three different HCC cell lines, also evaluating the impact of the drug alone and in association with sorafenib on proliferation and apoptosis induction.

Cell viability of HepG2, Hep3B and HuH7 was evaluated with the MTT assay after exposure to different concentrations of metformin (0–20 mmol/mL) and sorafenib (2.5 and 5 μmol/mL) for 48 h. A dose-dependent significant inhibition of cell viability was observed after metformin alone and in combination with sorafenib in all cell lines, as reported in Figure 3A–C. The combined treatment induced a greater arrest of cell growth than that observed after single agent exposure (Figure 3A–C).

To determine whether apoptosis contributed to the antiproliferative effect of both drugs, we examined phosphatidylserine externalization in cells after metformin at 20 mmol/mL and sorafenib at 2.5 μmol/mL for 48 h. The AnnexinV/PI assay showed a statistically significant increase in apoptosis after combination therapy compared to single drug administration (Figure 3D).

The protein expression levels of SIRT-3, p-mTOR and related-effectors were evaluated after metformin at 20 mmol/mL and sorafenib at 2.5 μmol/mL for 48 h (Figure 3E). In HepG2 cells, we observed no significant changes in the expression of p-AMPK after metformin alone and in combination with sorafenib. The single agent and the combined treatment down-regulated SIRT-3 and p-mTOR. In Hep3B, we observed an up-regulation in p-AMPK expression after metformin alone and in combination with sorafenib, whereas SIRT-3 remained unchanged and p-mTOR resulted up-regulated after metformin alone and in combined treatment. In HuH7, we observed no significant changes in the expression of p-AMPK and p-mTOR after metformin alone and in combination with sorafenib. SIRT-3 resulted down-regulated after the treatment with metformin alone and in combination with sorafenib.

### 2.4. Association between SIRT-3, p-mTOR and Nuclear HIF-1α Expression and Clinical Outcome of Early-Stage HCC Patients 

We further investigated the relationship between SIRT-3, p-mTOR and nuclear HIF-1α expression with clinical outcome of early-stage HCC patients (Table 2). With regards to SIRT-3, a slight, albeit not significant, benefit was observed for patients with higher expression.

Patients with HIF-1α tumor expression <1% showed a five-year percent DFS of 20 months (95% CI 80-44) and an eight-year percent OS of 0 months (95% CI), compared with a five-year percent DFS of 100 months (95% CI) and an eight-year percent OS of 50 months (95% CI 0-100) for patients with HIF-1α tumor expression ≥1% (*p* = 0.002; *p* = 0.041, respectively) (Table 2 and Figure 4).

No significant correlation was observed for p-mTOR in this setting of patients.

Moreover, in light to the positive intensity variations described in Appendix A for SIRT-3 and p-mTOR, we performed other analyses through two statistical methods reported in the literature [16,17], confirming our results.

### 2.5. High p-mTOR Levels Correlated with Poor Prognosis in Advanced-Stage HCC Patients

We correlated the expression levels of the markers with the response rate to sorafenib in advanced-stage HCC patients. As reported in Table 3, there was no significant difference observed.

Regarding the outcome of patients, we observed that advanced-stage HCC patients, with a median value of p-mTOR = 0, showed a median PFS of 5.3 months (95% CI 2.3–10.7) and a median OS of 13.9 months (95% CI 6.7–15.8), whereas those with a median value of p-mTOR > 0 had a median PFS of 1.8 months (95% CI 1.6–4.0) and a median OS of 6.1 months (95% CI 2.6–nr), (*p* = 0.055, *p* = 0.098, respectively) (Table 4). No significant correlation was observed for SIRT-3 and HIF-1α in this setting of patients. As above, we confirmed our statistical analyses considering the positive intensity variations for SIRT-3 and p-mTOR described in Appendix A.

## 3. Discussion

SIRT-3 is a member of the mitochondrial sirtuin family, involved in the regulation of different mechanisms including metabolism, aging, and cancer [18,19]. SIRT-3 acts as modulator of fatty-acid oxidation [20], at the same time promoting the urea cycle, by detoxifying the ammonia generated during amino acid catabolism [21]. In diabetic patients, SIRT-3 promotes the hepatic production of ketone bodies that are used by the brain and skeletal muscle as main energy sources [22], starting from fatty acid oxidation. Casadei Gardini et al. demonstrated that HCC patients in chronic treatment with metformin for T2DM show a lower response to sorafenib compared to non-diabetic patients or those taking insulin [5]. On the contrary, the in vitro study revealed cytotoxic effects of metformin and synergistic antitumor effects of sorafenib in our HCC cell lines. The discrepancy of these pre-clinical results with our clinical studies published previously [4,5] are due to the difficulty of reproducing in vitro the metabolic condition of HCC patients and, consequently, the effect of a chronic treatment with metformin. Recently, the same group confirmed these data in a large patient population, suggesting an involvement of SIRT-3 in the resistance mechanism to sorafenib in patients on chronic metformin therapy [4].

SIRT-3 represents a critical effector in the AMPK/mTOR pathway [6,8]. It is known that metformin activates AMPK, which, in turn, positively regulates SIRT-3 [23]. In the present work, we confirmed that SIRT-3 protein expression levels were significantly higher in patients with diabetes than those without diabetes, in a larger study population. In addition, the protein was more expressed in HCC patients treated with metformin than in those taking insulin, suggesting that SIRT-3 could be regulated by metformin-induced activation of AMPK.

Another important aspect is the expression of p-mTOR, a SIRT-3 downstream target. Although metformin has been shown to inhibit mTOR signaling [6,23,24], we observed that HCC patients with T2DM and those treated with metformin had higher levels of p-mTOR. Such upregulation suggests a mechanism of resistance to the effect of metformin. T2DM is caused by insulin resistance (IR), characterized by liver deficiency, adipose tissues, and skeletal muscles [25]. Recent studies have shown that IR is closely related to defective autophagy inhibited by mTOR [26,27,28]. Given that autophagy induction contrasts IR under diabetic conditions, Wan Zhou et al. hypothesized that rapamycin, a specific mTOR inhibitor, could promote autophagy to inhibit IR in T2DM [29].

Interestingly, we showed that in HCC patients with metabolic syndrome p-mTOR resulted more expressed than those with other etiologies. These data highlight the key role of p-mTOR in HCC metabolism and suggest this marker as a promising therapeutic target in HCC patients with metabolic dysfunctions. The detailed mechanisms linking autophagy and metabolic dysfunction in HCC remain an open question and it is necessary to improve our understanding with further investigations.

In light to our results that show the association of SIRT-3 and p-mTOR with metabolic etiology and presence of T2DM, we verified whether the expression of the markers was related to the presence of HCC or metabolic dysfunctions. In the cases where non-cancerous adjacent liver tissue was present, we showed an absent/slight positivity of SIRT-3 and a negative expression of p-mTOR, suggesting that the presence of tumor strongly affected SIRT-3 and p-mTOR expression.

To better explain how SIRT-3 regulates p-mTOR in a mechanism of drug resistance suggested in the previous ex vivo study [4], we analyzed the in vitro effect of metformin alone and in combination with sorafenib. We described different mechanisms in the three cell lines, suggesting a regulation of p-mTOR by p-AMPK/SIRT-3-independent mechanism after metformin and combined treatment. Indeed, in HepG2 cell line, we observed a downregulation of p-mTOR, a reduction in SIRT-3 levels, and no changes in p-AMPK expression. In Hep3B, the increased p-mTOR expression and the up-regulation of p-AMPK were described. In HuH7, we did not find a modulation of p-mTOR and p-AMPK, but only reduced SIRT-3 expression levels, more evident after the combined treatment.

The discrepancy of results in the three cell lines can be due to their different origin and molecular background. HuH7 cells are characterized by a stem-like phenotype and a more pronounced constitutive activation of the mTOR pathway than Hep3B (virus-related) and HepG2 (not virus-related). Secondly, the protein expression data obtained in vitro contrast those obtained ex vivo, probably because SIRT-3 and its downstream effectors have been investigated in patients undergoing chronic treatment with metformin for the presence of T2DM, which represents a complex clinical condition not reproducible in cell lines. Further studies are needed to improve our understanding of the role of these markers. Thirdly, it is still controversial whether the three markers act as tumor promoters or tumor suppressors in HCC [3], as literature data are supportive of a tumor suppressing function of SIRT-3 [12,13] and a tumor promoting function of p-mTOR and HIF-1α in HCC [14,15].

We separated the two cohorts of patients to clarify clinical significance of SIRT-3 and its downstream targets.

Our results describe a slight, albeit not significant, benefit for early-stage HCC patients with a high expression of SIRT-3 in line with Zhang and colleagues [12], and a significant benefit for those with a high nuclear HIF-1α expression contrary to what is reported in the literature [14,30,31]. The data relative to the prognostic role of HIF-1α surprised us, being in contrast with the oncogenic role described to date. Our study was performed on a small number of early-stage cases treated with surgery and a validation is necessary on a large cohort of patients. Further investigations could highlight a multifaceted role of HIF-1 in HCC at early stage or disprove our observation.

In advanced-stage HCC patients, a favorable trend of PFS and OS for patients with low expression of p-mTOR suggested this marker as a promising therapeutic target for HCC treatment, in line with other studies [15,32,33]. Our observations agree with literature data proving that the activation of the mTOR pathway in HCC is associated with bad prognosis and earlier recurrence [15,33]. Ponziani F et al. suggested a new therapeutic strategy implying the use of mTOR inhibitors in a combined pharmacological approach to improve HCC molecular-targeted therapy [32]. Based on this evidence, blocking this pathway could represent an attractive strategy for the treatment of HCC patients at an advanced stage.

However, since the potential mechanisms of SIRT-3, p-mTOR and HIF-1α in HCC are still unclear, their expression should be analyzed in a validation cohort.

In our study, despite being a retrospective evaluation, cases were selected consecutively to minimize bias. The discrepancy in the number of samples analyzed for all markers was related to the lack of sufficient biological material for all selected patients.

Our study highlighted an involvement of SIRT-3 and p-mTOR in metabolic dysfunctions that occur in HCC patients. Moreover, we suggested that, in the mechanism of resistance to sorafenib hypothesized in the previous works [4,5], p-mTOR activation in HCC patients undergoing chronic treatment with metformin for T2DM had a key role.

Regarding the clinical implication of three markers, we suggested SIRT-3 and HIF-1α as predictors of prognosis in early-stage HCC patients, and p-mTOR as a promising therapeutic target for the treatment of advanced-stage HCC.

## 4. Materials and Methods

### 4.1. Patient Population

The study population included 70 HCC patients: 48 early-stage cases (BCLC 0 or A) were enrolled at the Department of General Surgery, Morgagni-Pierantoni Hospital (AUSL Romagna, Forlì, Italy) and at Campus Bio-Medico University (Rome, Italy) between 2001 and 2016, and 22 advanced- or intermediate-stage cases [either histologically proven or diagnosed according to the AASLD (American Association for the Study of Liver Diseases 2005) guidelines] refractory or no longer amenable to locoregional therapies were enrolled at the Istituto Scientifico Romagnolo per lo Studio e la Cura dei Tumori (IRST) IRCCS (Meldola, Italy) between 2008 and 2016. Biopsy samples from advanced- or intermediate-stage patients were taken 6 months before patients received sorafenib therapy. This study was approved by the local Ethics Committee (Prot. 7523/2015—IRSTB050) in accordance with the principles laid down by the Helsinki Declaration. Informed consent was obtained from each patient included in the study.

### 4.2. Immunohistochemistry Analysis

Tumor samples were fixed in 10% formalin. Four-micrometer sections from paraffin- embedded block tissue sections were mounted on positive-charged slides (BioOptica, Milan, Italy). The expression of SIRT-3, p-mTOR and HIF-1α was measured by immunohistochemistry using the Ventana Benchmark XT staining system (Ventana Medical Systems, Tucson, AZ, USA), with Optiview DAB Detection Kit (Ventana Medical Systems). Tissue sections were incubated for 1 h with a rabbit monoclonal antibody directed against SIRT-3 (Cell Signaling, Leiden, The Netherlands: Clone C73E3) diluted 1:100, p-mTOR (Ser2448) (Elabscience, Houston, TX, USA) diluted 1:50 and HIF-1α (Abcam, UK: Clone EP1215Y) diluted 1:300. To reduce the background, the antibodies were diluted with an antibody diluent with casein (Ventana Medical Systems, Tucson, AZ, USA), a buffered solution containing salt and immunoglobulins. Immunoreactivity was expressed as the ratio between the percentage of immunopositive cells and the entire area of invasive neoplastic tissue. Each tissue was also evaluated by staining intensity (0, absent; 1, weak; 2, moderate; 3, strong staining). HepG2 cell line, breast cancer tissue and a colon adenocarcinoma tissue were used as positive controls for SIRT-3, p-mTOR and HIF-1α, respectively.

### 4.3. HCC Cell Lines and Drugs

Human HepG2 and Hep3B cell lines were obtained from the American Type Culture Collection (Rockville, MD, USA), whereas the HuH7 cell line was kindly supplied by the Center for Applied Biomedical Research (St. Orsola-Malpighi Hospital, University of Bologna, Bologna, Italy). Cell lines were cultured in standard DMEM supplemented with 10% fetal bovine serum (FBS) (GE Healthcare, Piscataway, NJ, USA), 2 mM l-glutamine (EuroClone, UK), penicillin (100 U/mL) and streptomycin (100 μg/mL) (GIBCO, UK) and kept at 37 °C in a humidified incubator with 5% CO_2_. Cells were used in the exponential growth phase in all experiments and checked periodically for mycoplasma contamination by MycoAlert™ Mycoplasma Detection Kit (Lonza, Basel, Switzerland). Cells were seeded at a density of 100,000 cells/cm^2^. Cells were treated with Sorafenib and Metformin, purchased from Santa Cruz and Sigma Aldrich, respectively. Drugs stocks were freshly diluted in culture medium before each experiment.

### 4.4. Cell Viability Assay

Cell viability was determined by 3-(4,5-dimethylthiazol-2-yl)-2,5-diphenyltetrazolium bromide (MTT) (Sigma Aldrich). At the end of treatment, cells were incubated with 1 mg/mL MTT, prepared in supplemented DMEM culture medium, for 2 h at 37 °C. Cells were then washed with phosphate buffer saline (PBS) (0.2 M, pH 7.4) and the reduced MTT formazan crystals were solubilized with dimethyl sulfoxide (DMSO) (Carlo Erba, Milano, Italy). The optic density was determined at a wavelength of 570 nm by a colorimetric plate reader.

### 4.5. Annexin V Assay

Cells (1 × 10^6^ cells/sample) were washed once in PBS and incubated with 25 µL/mL Annexin V-FITC in binding buffer (eBioscence) for 15 min at 37 °C in a humidified atmosphere in the dark. Cells were then washed in PBS and re-suspended in binding buffer. Immediately before flow cytometric analysis, propidium iodide (PI) was added to a final concentration of 5 µg/mL.

### 4.6. Western Blot Analysis

Total proteins were extracted with 1× RIPA lysis buffer (Santa Cruz Biotechnology, Santa Cruz, CA, USA) with the addition of 10 µL PMSF, 10 µL sodium orthovanadate and 15 µL protease inhibitors, per mL of 1× RIPA lysis buffer. Proteins were quantified using BCA Protein Assay (Pierce, ThermoScientific, Madison, WI, USA) and with a Multiscan EX microplate reader (Thermo Labsystems, Helsinki, Finland). Proteins (amount: 40 µg for p-AMPK and p-mTOR, and 20 µg for SIRT-3) were denatured and separated by electrophoresis using a Criterion TGX Gel Precast 4–20% (Bio-Rad Laboratories, Hercules, CA, USA) and Laemmli Sample Buffer (Bio-Rad Laboratories) with 5% β-mercaptoethanol (Carlo Erba Reagents, Milan, Italy). Electrophoretic run was performed at 180 V in a TRIS/Glycine/SDS 1× buffer (Bio-Rad). Proteins were then transferred on a PVDF membrane (Trans-Blot Transfer Turbo midi-format 0.2 µm; Bio-Rad Laboratories) using the Trans Blot Turbo System (Bio-Rad Laboratories). The membrane was incubated at room temperature in a solution of Tween 20 (Bio-Rad Laboratories) at 0.1% and 1× Dulbecco’s Phosphate Buffered Saline (PBS; Gibco) supplemented with 5% milk powder (Blotting Grade Blocker Non Fat Dry Milk; Bio-Rad Laboratories). We used the following primary antibodies and dilutions: p-AMPK (Thr172) (D4D6D, 1:1000), SIRT-3 (C73E3, 1:1000) and p-mTOR (Ser2448) (clone D9C2, 1:1000) (Cell Signaling), Vinculin (clone VLN01) (Thermo Scientific, 1:1000). Secondary antibodies and dilutions used are the following: goat anti-rabbit and anti-mouse IgG-HRP (Santa Cruz Biotechnology, 1:5000), and StrepTactin-HRP Conjugate (Bio-Rad 1:10000). Images were acquired using Clarity Western ECL Substrate (Bio-Rad Laboratories) and Chemidoc (Bio-Rad Laboratories), and analyzed using ImageJ Software.

### 4.7. Statistical Analysis 

The expression levels of SIRT-3, p-mTOR and HIF-1α were summarized using descriptive statistics such as absolute and relative frequency, means, standard deviation, median and interquartile range (IQR). The Wilcoxon test was performed to compare continuous variables and Chi-Square or Fisher’s exact test were performed to compare categorical variables, as appropriate. The cut-offs for markers were determined through median value. Disease-Free Survival (DFS) was defined as the time elapsed between the date of start of therapy and the date of first disease progression or last follow-up. Progression-Free Survival (PFS) was defined as the time elapsed between the date of start of therapy and date of disease progression or last tumor evaluation. Overall Survival (OS) was defined as the time elapsed between the date of start of therapy and date of death for any cause or the date of last follow-up. Event-time distributions were estimated by the Kaplan–Meier method and were compared using the log-rank test. Statistical analyses were performed with SAS software version 9.4 (SAS Institute, Cary, NC, USA).

In vitro data are presented as the mean ± standard deviation (SD) from at least three biological replicates. Statistical analysis was performed with Graphpad (GraphPad Prism 6).

All *p*-values < 0.05 were considered as significant.

## Figures and Tables

**Figure 1 ijms-20-01503-f001:**
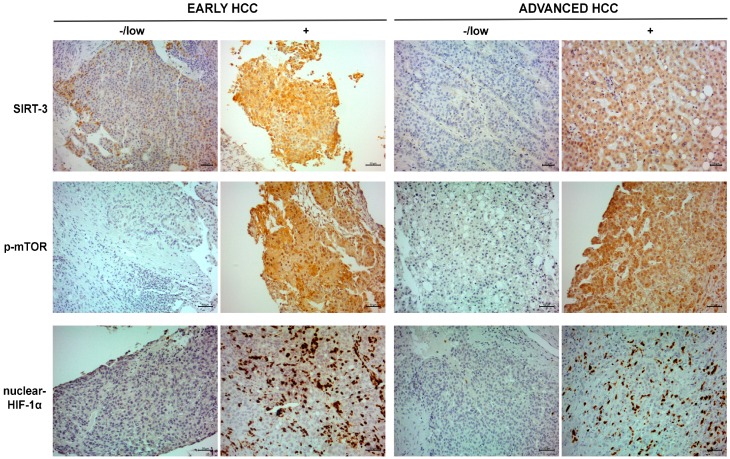
Immunohistochemistry for SIRT-3, p-mTOR and HIF-1α. Representative cases related to positive SIRT-3, p-mTOR and nuclear HIF-1α expression in early-stage (**left**) and advanced-stage HCC patients (**right**). Scale bar: 20 µm and 50 µm.

**Figure 2 ijms-20-01503-f002:**
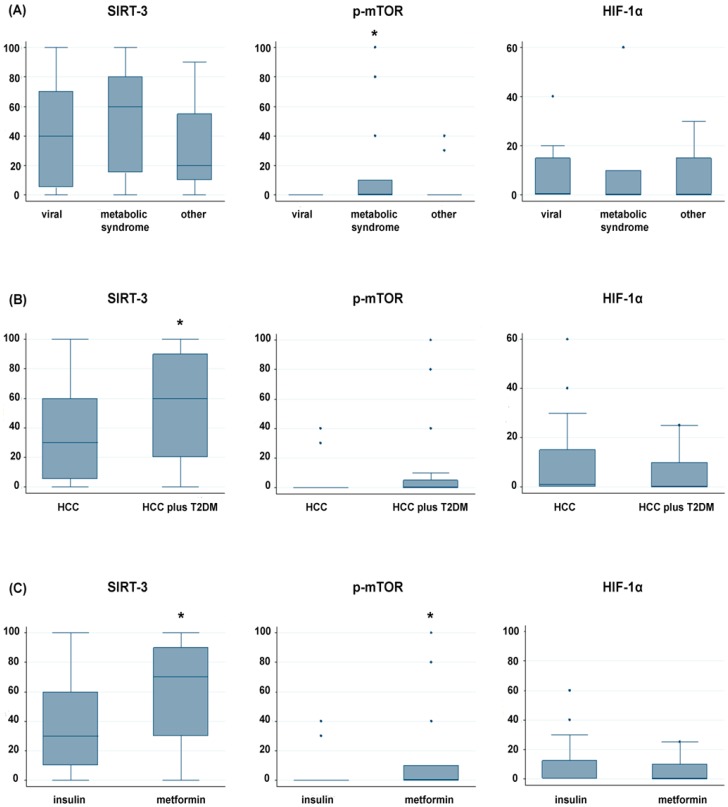
Representative box plots of expression of SIRT-3, p-mTOR and HIF-1α in relation with: (**A**) etiology (Metabolic syndrome vs. all; *, *p* < 0.05); (**B**) the presence of T2DM (*, *p* < 0.05); and (**C**) therapy with insulin or metformin (*, *p* < 0.05).

**Figure 3 ijms-20-01503-f003:**
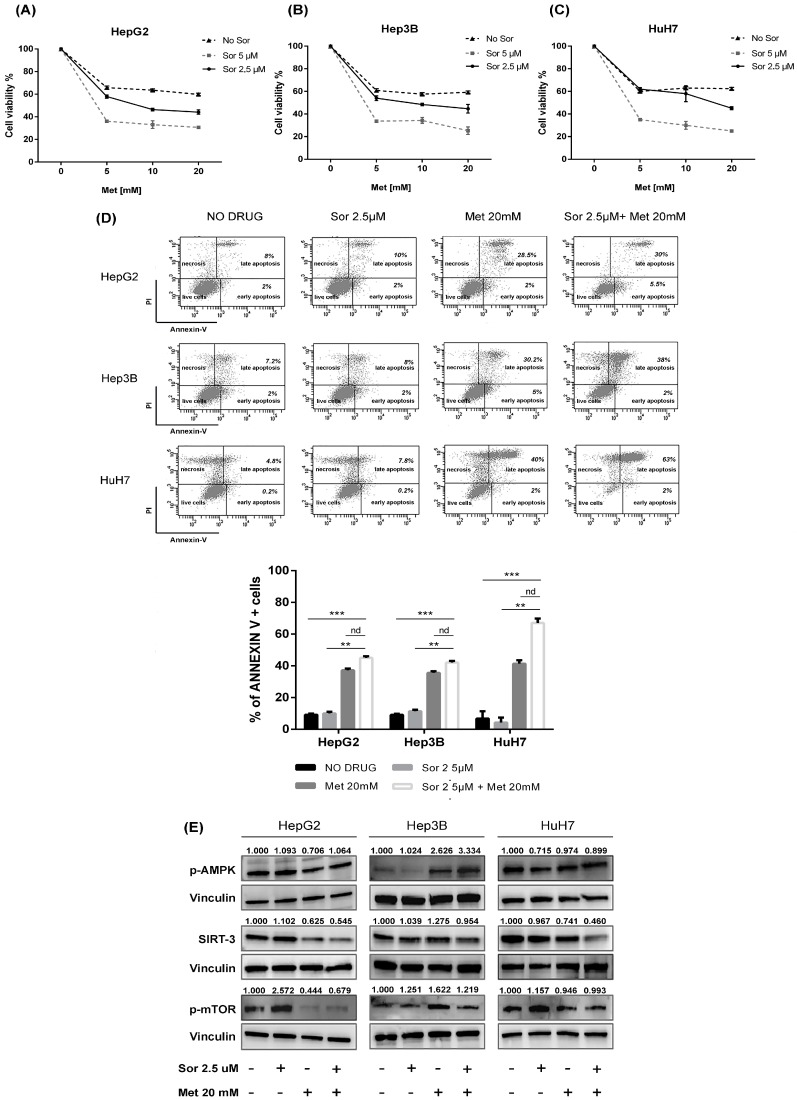
The in vitro effect of metformin and sorafenib on HCC cell lines. MTT assay for cell survival assessment in three HCC cell lines: (**A**) HepG2; (**B**) Hep3B; and (**C**) HuH7, before and after treatment with metformin (Met) [0–20 mmol/mL] alone and in combination with sorafenib (Sor) [2.5 and 5 µmol/mL] for 48 h. (**D**) Dot plots and relative quantification of annexin V+ cells (early and late apoptosis) in HCC cell lines treated with DMSO (NO DRUG), Met at 20 mmol/mL and Sor at 2.5 µmol/mL used alone and in combination for 48 h. For all experiments, values represent the mean ± SD of three biological replicates (** *p* < 0.01, *** *p* < 0.001). (**E**) Representative immunoblots showing the expression of p-AMPK, SIRT-3 and p-mTOR and relative quantified values of the bands after treatment with Met 20 mM and Sor 2.5 µM alone and in combination for 48 h in HepG2, Hep3B and HuH7 cell lines. Vinculin was used as loading control.

**Figure 4 ijms-20-01503-f004:**
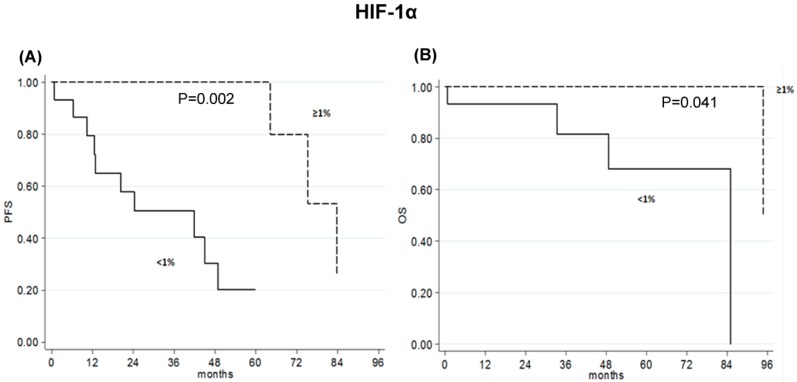
Kaplan–Meyer curves. (**A**) DFS and (**B**) OS relative to nuclear HIF-1α expression < or ≥1% in early-stage HCC patients.

**Table 1 ijms-20-01503-t001:** Clinical characteristics of HCC patients.

Patients	N. (%)
Median age (range)	70 (28–89)
Gender	
Male	60 (85.7)
Female	10 (14.3)
Etiology	
Viral infection	31 (44.3)
Metabolic syndrome	23 (32.9)
Alcoholic	7 (10.0)
Other	9 (8.6)
Diabetes	
No	38 (55.1)
Yes	31 (44.9)
Metformin	
No	46 (66.7)
Yes	23 (33.3)
Performance Status (ECOG)	
0	56 (80)
1	14 (20)
Child–Pugh	
A	67 (89.7)
B	3 (10.3)
BCLC staging	
A	41 (58.6)
B	9 (12.9)
C	20 (28.5)
Alfafetoprotein levels	
<400	17 (63.0)
≥400	10 (37.0)
Unknown	43
MELD score	
≤10	18 (25.7)
>10	52 (74.3)
Metastasis	
No	56 (80)
Yes	14 (20)

**Table 2 ijms-20-01503-t002:** Expression of markers in relation to clinical outcome in early-stage HCC patients.

Markers	No. Patients	Five-Year Percent DFS (months) (95% CI)	*p*	Eight-Year Percent OS (months) (95% CI)	*p*
SIRT-3 (%)					
<35	23	23 (4–41)		20 (0–49)	
≥35	23	51 (28–73)	0.308	32 (0–78)	0.117
p-mTor (%)					
0	23	47 (22–73)		39 (0–95)	
>0	4	50 (1–99)	0.952	0	0.577
HIF-1α (%)					
<1	15	20 (0–44)		0	
≥1	10	100	0.002	50 (0-100)	0.041

DFS, disease-free survival; OS, overall survival.

**Table 3 ijms-20-01503-t003:** Correlation between the SIRT-3, p-mTOR and HIF-1α expression levels and response rate to sorafenib in advanced-stage HCC patients.

Response	No. Patients	SIRT-3 (%) Median Value (range)	*p*	No. Patients	p-mTOR (%) Median value (range)	*p*	No. Patients	HIF-1α (%) Median Value (range)	*p*
CR	4	80 (60–100)		2	0 (0–0)		2	20 (0–40)	
SD	10	45 (0–100)		7	0 (0–0)		7	0 (0–8)	
PD	10	55 (0–90)	0.276	5	0 (0–80)	0.407	4	0 (0–60)	0.752

**Table 4 ijms-20-01503-t004:** Expression of markers in relation to clinical outcome in advanced-stage HCC patients.

Markers	No. Patients	Median PFS (months) (95% CI)	*p*	Median OS (months) (95% CI)	*p*
SIRT-3 (%)					
<60	11	3.7 (1.6–6.0)		12.0 (5.2–nr)	
≥60	10	5.3 (1.7–12.9)	0.542	14.3 (2.0–15.8)	0.624
p-mTor (%)					
0	13	5.3 (2.3–10.7)		13.9 (6.7–15.8)	
>0	3	1.8 (1.6–4.0)	0.055	6.1 (2.6–nr)	0.098
HIF-1α (%)					
0	10	3.5 (1.2–6.0)		13.9 (2.2–15.8)	
>0	6	4.7 (1.6–22.8)	0.382	12.0 (2.6–22.8)	0.682

nr: not reached; PFS: progression-free survival; OS: overall survival.

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
